# WHAT MAKES PRE-TRAINED VISUAL REPRESENTATIONS SUCCESSFUL FOR ROBUST MANIPULATION?

## ABSTRACT

Inspired by the success of transfer learning in computer vision, roboticists have investigated visual pre-training as a means to improve the learning efficiency and generalization ability of policies learned from pixels. To that end, past work has favored large object interaction datasets, such as first-person videos of humans completing diverse tasks, in pursuit of manipulation-relevant features. Although this approach improves the efficiency of policy learning, it remains unclear how reliable these representations are in the presence of distribution shifts that arise commonly in robotic applications. Surprisingly, we find that visual representations designed for manipulation and control tasks do not necessarily generalize under subtle changes in lighting and scene texture or the introduction of distractor objects. To understand what properties *do* lead to robust representations, we compare the performance of 15 pre-trained vision models under different visual appearances. We find that emergent segmentation ability is a strong predictor of out-of-distribution generalization among ViT models. The rank order induced by this metric is more predictive than metrics that have previously guided generalization research within computer vision and machine learning, such as downstream ImageNet accuracy, in-domain accuracy, or shape-bias as evaluated by cue-conflict performance. We test this finding extensively on a suite of distribution shifts in ten tasks across two simulated manipulation environments. On the ALOHA setup, segmentation score predicts real-world performance after offline training with 50 demonstrations.

## 1 INTRODUCTION

In spite of vast progress in computer vision, the question of how to learn a good visual representation for robotics remains open (Chen* et al., 2021). Elsewhere in computer vision, internet datasets are retrofit to new tasks with transfer learning, which promises both generalization and fast adaptation to downstream tasks in exchange for large-scale pre-training. But in the field of robotics, this promise has yet to be fulfilled even though policies learned from pixels struggle substantially with data efficiency (Cobbe et al., 2018) and especially generalization under visual changes in a scene (Cobbe et al., 2019a).

Recent work (Damen et al., 2018; Grauman et al., 2022) posits that the missing piece is a large pre-training dataset of object interactions across diverse environments — the ImageNet (Deng et al., 2009) or CommonCrawl (Raffel et al., 2020) of manipulation. That is, if we want to improve the visual generalization ability of pre-trained models we simply need to collect datasets of this kind at scale. Indeed, training on large datasets of first-person human interaction data increases policy performance and learning efficiency downstream (Nair et al., 2022; Xiao et al., 2022), but these evaluations occur in environments that are very similar to those used for policy learning. Robotic applications commonly contain environments with varying lighting conditions, scene textures, and background objects, and we want pre-trained representations to allow the robot to handle such variability. Yet we have few concrete measures of how well pre-trained representations generalize out-of-distribution. To take a step towards understanding these problems, our goal in this paper is to thoroughly answer the questions *"which models generalize?"* and *"how can we predict how well a pre-trained model will generalize?"*

**Our first key finding** is that, when evaluated under visual distribution shifts, models that are designed for manipulation and control do not outperform standard visual pre-training methods. This finding violates our intuitions about what is needed to scale up robot learning and brings into question what constitutes relevant data, how to quantify useful features, and the importance of design choices such as model architecture. In other words, we need more guiding principles to help us understand what representations are good for manipulation and make the problem of iterating on pre-training strategies much more straightforward. Currently, evaluating a pre-trained policy requires training and rolling out downstream policies across multiple environments and experimental conditions. Instead, we can take inspiration from computer vision, which has developed proxies for robust performance on vast out-of-distribution datasets (Geirhos et al., 2021).

**Our second key finding** is that the emergent segmentation ability of a ViT model is a strong predictor of out-of-distribution generalization performance. We visualize this phenomenon, which we refer to as "spatial features," in Figure 1. Other metrics of model quality, such as linear probes on ImageNet (Chen et al., 2020), and metrics of out-of-distribution performance, such as in-domain accuracy (Miller et al., 2021) and shape-bias (Geirhos et al., 2019), are not predictive for this model class, despite their predictive power in other commonly-studied domains like image classification. This hints at the possibility that the transfer setting of manipulation differs from computer vision tasks typically studied within the robustness literature.

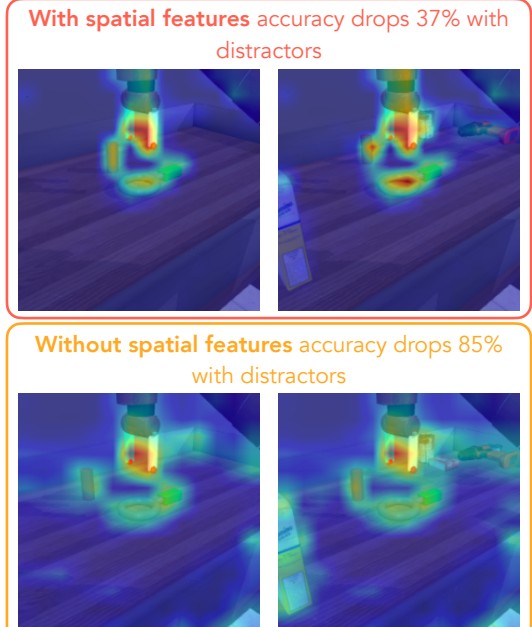

Figure 1: We find that the emergent segmentation ability of ViT attention heads (measured by Jaccard index) predicts performance under visual distribution shift. We refer to models with this property as having "spatial features." Notice how the attention of MVP shifts towards the sugar box distractor object in the bottom right image. The attention of DINO on the top shifts less. The impact of this factor overshadows other design choices such as data relevance.

To reach the conclusions above, we run 9,000 different simulated evaluations. Our simulated environments are adapted from two different existing visual distribution shift benchmarks (Xing et al., 2021; Xie* et al., 2023) to capture the shifts that arise commonly in robotics applications: changes in lighting, background and object texture, and the appearance of distractors. More specifically, we train policies on top of 15 pre-trained models, including 4 models designed for manipulation or control: R3M (Nair et al., 2022), two MVP variants (Xiao et al., 2022; Radosavovic et al., 2022), and VIP (Ma et al., 2022). We further validate these findings by comparing a model designed for manipulation against a model with a similar parameter count on a real-world screwdriver pick-up task using the ACT training framework (Zhao et al., 2023). Through these experiments, we make two striking findings: (1) pre-trained visual models designed for control do not necessarily generalize better than models pre-trained on more standard computer vision datasets and (2) the emergent segmentation performance of a ViT model is a strong predictor of the out-of-distribution generalization of a down-stream policy.

## 2 RELATED WORK

**Representation learning for manipulation.** The correct approach to visual representation learning for robotics is still an open question. There is evidence that separating visual representation learning from policy learning can further improve performance (Pari et al., 2022; Parisi et al., 2022). Recent works have shown that models pre-trained on large manipulation-relevant datasets (Goyal et al., 2017; Damen et al., 2018; Shan et al., 2020; Grauman et al., 2022) or learned with visual affordances

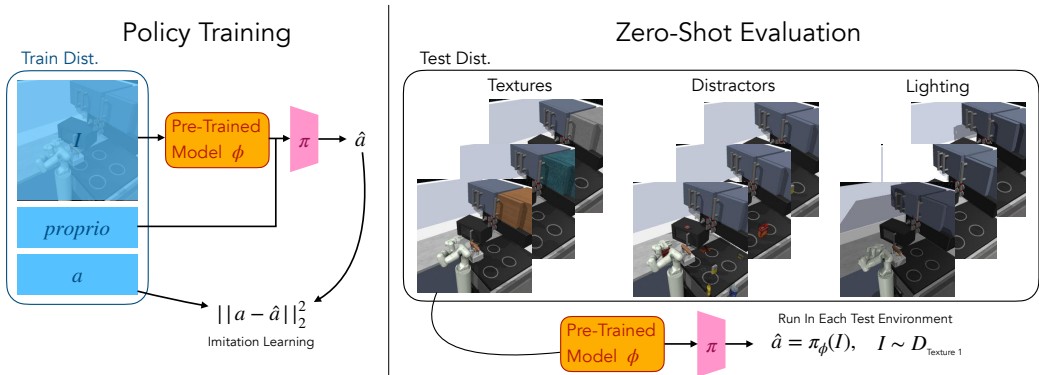

Figure 2: **Evaluation Scheme.** We begin our evaluation procedure by training a policy with behavior cloning on top of frozen features. In every experimental setting, we ablate the encoder used to extract features from the image observation. The learned policy is then evaluated in each of the visual shift environments to attain a zero-shot success value.

from RGBD data (Yen-Chen et al., 2020) can improve the efficiency and performance of policy learning (Karamcheti et al., 2023) in comparison to standard vision datasets such as ImageNet (Deng et al., 2009), but they do not focus on performance under visual distribution shift. We evaluate the performance of R3M (Nair et al., 2022), MVP (Xiao et al., 2022; Radosavovic et al., 2022), and VIP (Ma et al., 2022). Other work has studied generalization of pre-trained representations to new reinforcement learning tasks for manipulation (Ma et al., 2022) and navigation (Sax et al., 2018) where the agent is able to train on visual data from the new environment. Separate from the question of pre-training visual representations is the question of how to best train policies on top of pixel observations (Laskin et al., 2020b; Yarats et al., 2021). Majumdar et al. (2023) benchmarks the performance of pre-trained visual representations on a handful of manipulation environments, but they focus on in-domain performance and also investigate navigation environments. Hu et al. (2023) shows that model performance is highly sensitive to downstream policy learning strategy. We use imitation learning for our evaluation protocol, which they find to be a more stable measure of performance.

**Robustness in computer vision.** There is extensive work studying the impact of design choices, such as architecture, loss, and data, on the performance of visual models under distribution shift. See Geirhos et al. (2021) for a comprehensive comparison. Most relevant to our paper are studies of shape-bias and architecture. While shape-biased models tend to be more robust than texture-biased ones (Geirhos et al., 2019), the impact of architecture on robustness is less straightforward. For example, vision transformers exhibit better robustness to universal adversarial attacks (Shao et al., 2022), but they are more susceptible to patch-level attacks (Fu et al., 2022). When compared on natural distribution shifts (Hendrycks & Dietterich, 2019; Hendrycks et al., 2021a;b), vision transformers and convolutional networks achieve comparable performance when provided with enough data (Bhojanapalli et al., 2021). But for occlusions specifically, vision transformers appear to have an edge (Naseer et al., 2021). Miller et al. (2021) studies the predictive power of in-domain performance for out-of-distribution generalization. Unlike all of these prior works, we focus on how pre-trained representations affect robustness in downstream robotics tasks, instead of downstream vision tasks.

**Learning robust policies.** Unlike work that focuses on changes in dynamics or initial state distribution (Huang et al., 2021; Raileanu et al., 2020; Laskin et al., 2020a; Cobbe et al., 2019b; Packer et al., 2018; Farebrother et al., 2018), we focus exclusively on the setting of visual distribution shifts. Kirk et al. (2021) and Zhao et al. (2019) provide a comprehensive survey on non-visual distribution shifts in decision making problems. Policy adaptation approaches enable visual robustness specifically by leveraging insights from domain adaptation during policy training (Hansen & Wang, 2021; Fan et al., 2021; Yoneda et al., 2021) or during deployment (Hansen et al., 2021). Other policy adaptation approaches blend pre-training together with reinforcement learning across diverse visual environments (Yuan et al., 2022). In the special case of closing the sim-to-real domain gap, a popular approach is to add randomized textures while training in simulation (Sadeghi & Levine, 2017; Tobin

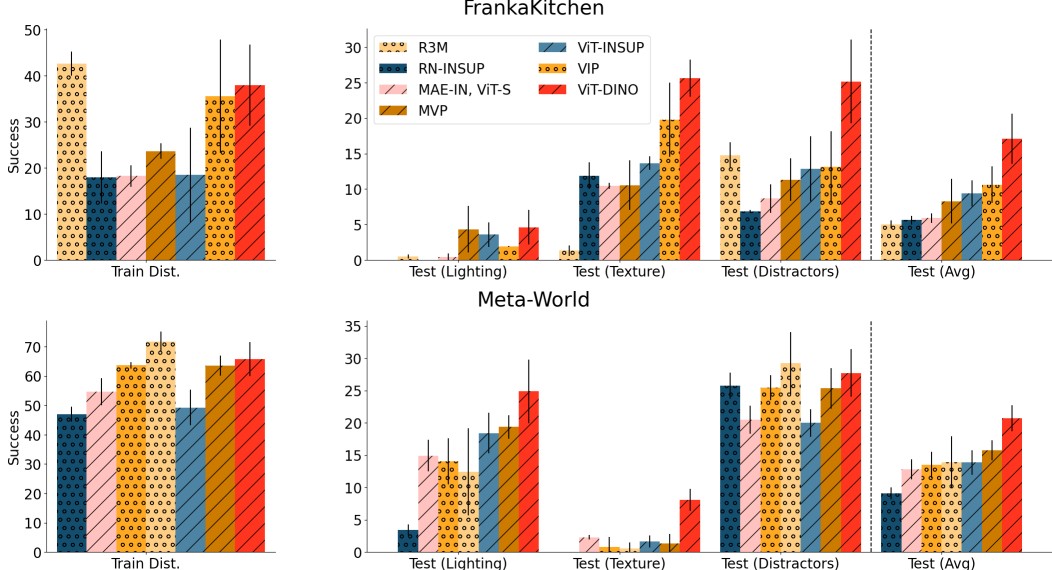

Figure 3: **Visual Generalization Performance.** Models trained with supervision on ImageNet are shades of blue. Models trained with self-supervision on ImageNet are in red. Models trained explicitly for manipulation and control tasks are orange. Dotted bars denote ResNets and slashed bars denote ViTs. Surprisingly, the best performing models are not necessarily the ones designed for manipulation. Each bar is an average over 30 experimental conditions.

et al., 2017; Peng et al., 2018; James et al., 2019). By contrast, our work is interested in explaining properties of a robust visual model for control. Consequently, our insights can be leveraged with or without any task specific data.

## 3 ENVIRONMENTS, EVALUATION PROTOCOL, AND PRE-TRAINED MODELS

Our goal is to understand how robust existing representations for manipulation are to visual distribution shifts that are realistic in robotic applications. To that end, we learn policies on top of frozen, pre-trained encoders and then evaluate these policies zero-shot under changes in lighting, object and scene texture, and the presence of distractors. These shifts are visualized in Appendix Figure 8 and a high level summary of our evaluation procedure is visualized in Figure 2. In this section, we describe the specifics of the manipulation environments, distribution shifts, and policy training setups.

**Environments and tasks.** We study ten tasks across two simulated manipulation environments, which are selected based on their popularity in studying learning-based approaches to manipulation. Within FrankaKitchen (Gupta et al., 2020) we evaluate performance on opening a microwave, sliding a cabinet door open, pulling a cabinet open, turning a knob, and turning on a light. Within Meta-World (Yu et al., 2019) we study assembling a ring onto a peg, placing an object between two bins, pushing a button, opening a drawer, and hammering a nail.

**Distribution shifts.** We construct environments to study out-of-distribution generalization within FrankaKitchen and Meta-World. Within FrankaKitchen, we reimplement the texture and lighting changes from KitchenShift (Xing et al., 2021). Within Meta-World we use texture changes from Xie* et al. (2023) and reimplement the same lighting changes as in FrankaKitchen. In both environments we include three levels of distractors: one, three, and nine YCB objects (Calli et al., 2015). We show average performance on each of these distributions shifts as well as performance on the original training distribution, which samples initial positions of the table and kitchen at random. More details about the implementation and parameterization of the distribution shifts are provided in Section A.3.

**Policy training.** Policy training is done in the same manner as R3M (Nair et al., 2022). A summary of the evaluation scheme is provided in Figure 2. We train an MLP on top of the pre-trained embedding with imitation learning (IL), which, given actions sampled from expert trajectories, $a \sim \mathcal{D}_{train}$,

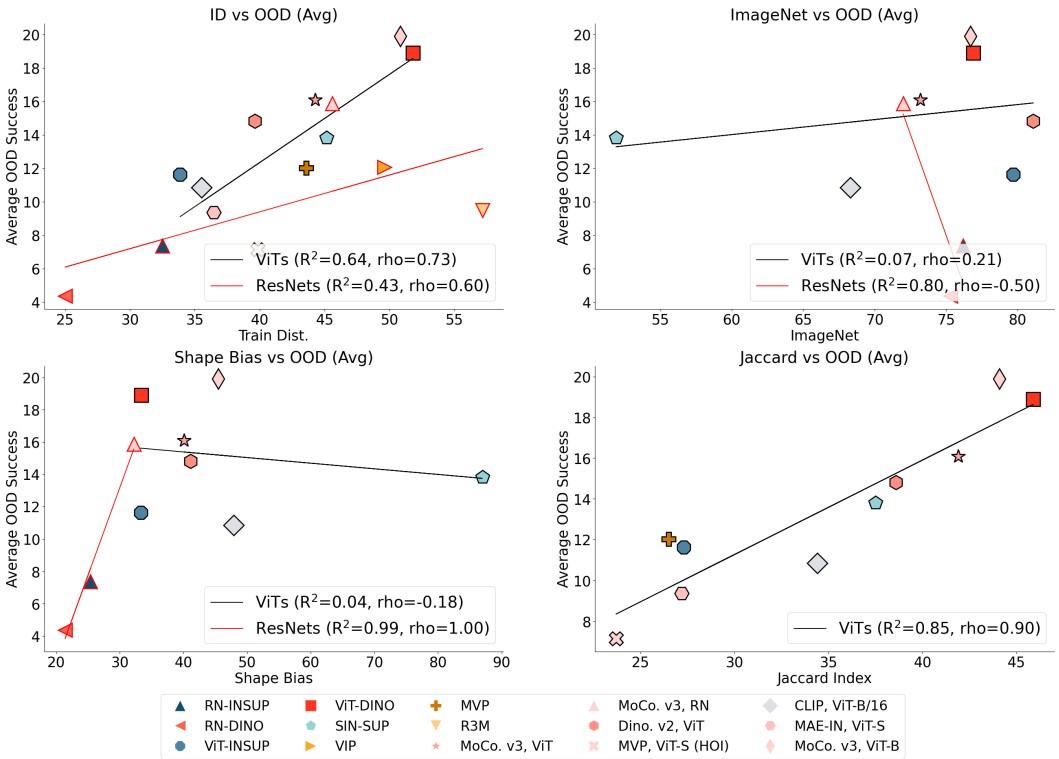

Figure 5: We plot the relationship between different metrics and out-of-distribution (OOD) generalization. There is a promising correlation between shape-bias and OOD performance for ResNets, but not ViTs. Instead, OOD performance for ViTs is strongly correlated with Jaccard index.

minimizes the mean squared error objective, $||a - \hat{a}||_2^2$. Here $\hat{a}$ denotes the action predicted from a given policy. Details of the training procedure are provided in Section A.4. The embedding weights are frozen during policy learning, so the pre-trained models receive no task data. We train 3 different seeds within each task for each of two different camera angles. In total, we learn 60 policies for each model and perform 11 evaluations per policy, including on the train distribution.

Formally, for a pre-trained representation $\phi$ we learn policies, $\pi_\phi$, each trained with a different seed, camera angle, and task. We average the performance of $\pi_\phi$ along each experimental condition and compute the mean performance and error across seeds.

**Models.** We categorize models by loss type and data source: supervised ImageNet models, self-supervised ImageNet models, and models trained for manipulation and control tasks.

## 4 GENERALIZATION OF MODELS PRE-TRAINED FOR MANIPULATION

One factor motivating work in learning-based robotics is the hypothesis of scale: if we collect more high-quality manipulation data, we should see improvements in policy generalization. However, our understanding of what high-quality data looks like for manipulation and control tasks is still imprecise. Past work on pre-training visual representations for manipulation and control tasks has focused on collecting large object interaction datasets and developing manipulation-relevant losses. But the generalization ability of such models in comparison to standard pre-training methods is still unknown. The goal of this section is to ask: *which models generalize?*

To focus our analysis, we compare models pre-trained for manipulation to two self-supervised ImageNet models and two supervised ImageNet models. Our main result is presented in Figure 3 where we plot the average success rate of the learned policies in the training environment distribution, within each class of visual shift, and across all types of visual shifts. All of the model names as well as the datasets, dataset sizes, model sizes, and loss functions are listed in Appendix Table 2.

We recommend that readers visit this table to get a high level view of each model in our comparison suite.

**Models pre-trained for manipulation.** Past work has trained visual representations for manipulation in two ways: by training with manipulation-specific losses or on data of human-object interactions. We focus on three recently introduced pre-trained models for manipulation that use different combinations of these approaches: Masked Visual Pretraining (MVP) (Xiao et al., 2022), Reusable Representations for Robot Manipulation (R3M) (Nair et al., 2022), and Value-Implicit Pre-Training (VIP) (Ma et al., 2022). We include important characteristics of these models, including dataset sizes, architecture sizes, and augmentations in Section A.1 and Table 2.

These models perform strongly within the training distribution: R3M and VIP in particular comfortably beat standard pre-training baselines. This is expected, especially for R3M which was evaluated on the same training environment. However, under subtle distribution shifts, models designed for manipulation struggle to generalize as well as supervised or self-supervised training with ImageNet. This is surprising for a few reasons. First, each manipulation model is trained on a larger dataset than the pre-trained baselines. Ego4D alone is 4.5M frames while ImageNet is only 1.2M. By parameter count, MVP is also larger than the ViT-S baselines. Finally, we expect human-object interaction datasets such as Ego4D to be more similar to the distribution of images observed when training a manipulation policy. The view-points are more varied and the scenes are less curated than ImageNet. Although we expect this to

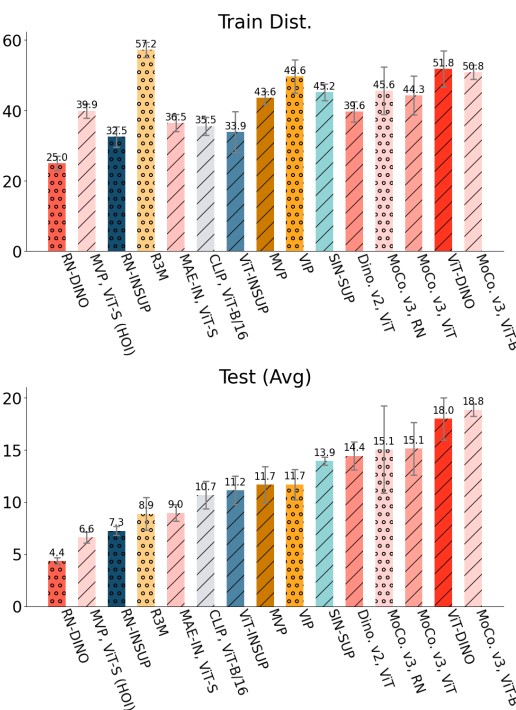

Figure 4: Average success rates for training and test distribution across both environments for every model in our evaluation suite. The best-performing model that was designed for manipulation ranks seventh out of all models evaluated.

improve the generalization of the learned policy, these results show that other factors may supersede the impact of data relevance or scale alone.

**Supervised ImageNet models.** Supervised training on ImageNet has long been a baseline for visual pre-training. Past work has found that features learned with supervised learning on ImageNet are also a strong baseline for control: even frozen features are competitive with ground-truth state information on a variety of simulated control tasks (Parisi et al., 2022). However, Parisi et al. (2022) also find that self-supervised learning outperforms supervised learning. Our results contradict this finding. Figure 4 shows that supervised training on Stylized ImageNet achieves a higher success rate in the training distribution than self-supervised training on ImageNet with a masked auto-encoding loss. These models maintain the same rank out-of-domain as well. Even without stylization, in-domain performance of supervised ImageNet models are competitive with models trained with MAE on FrankaKitchen. From these results, we conclude that the presence of supervision is not as predictive of in-domain or out-of-domain performance as other factors. We also find that supervised ImageNet training is still a strong baseline for model generalization: in both settings ViT-INSUP outperforms R3M and MVP.

**Self-Supervised ImageNet Models.** In Figure 3 we include two self-supervised ViT-S models. Under visual distribution shifts, the model trained with the DINO objective outperforms all three models that are designed for manipulation. Moreover, this trend holds for every distribution shift except Meta-World with distractors. The distractors evaluation suite averages over different levels of distractions and therefore favors models with a high performance in training. In Appendix Section A.8 we plot model performance across different levels of distractors and find that several self-supervised

ViTs experience a smaller drop in performance as more distractors are added compared to ResNet based pre-trained manipulation models like R3M and VIP.

Training with masked autoencoding performs well under distribution shifts in Meta-World, but is less strong under distribution shifts within FrankaKitchen. In Figure 4, we see that MoCo. v3, ViT-B also performs strongly out-of-distribution. When we compare MoCo and DINO against MAE-style training we see that MoCo and DINO use a more extensive set of augmentations. Taking this into account alongside the observation that a ViT trained with supervision on Stylized ImageNet performs well out-of-distribution we conclude that choice of augmentations outweighs the importance of supervision. This extends the findings of Geirhos et al. (2021) to the setting of robust manipulation.

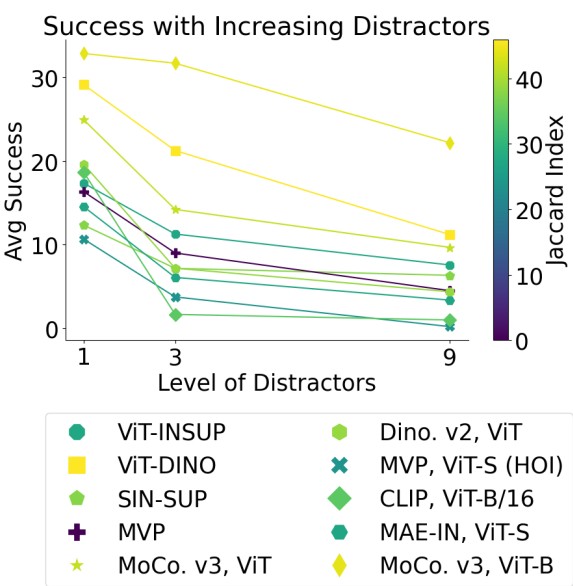

Figure 6: What happens to models with a high Jaccard index under an object-level distribution shift? Surprisingly, the models with the highest Jaccard index maintain the highest performance as the number of distractors increases.

**ViTs vs ResNets.** One important design choice when selecting a pre-trained model is the choice of architecture. We focus on ResNets and ViTs. In all of our experiments, we use ResNet-50 (He et al., 2016) to be consistent with past work on visual pre-training (Parisi et al., 2022; Nair et al., 2022; Ma et al., 2022). Vision transformers (ViTs) (Dosovitskiy et al., 2021) have seen widespread adoption within computer vision (Khan et al., 2022), but have only recently been used for learning representations for control (Xiao et al., 2022). We find that, on average, ViTs have a slight edge on out-of-distribution generalization compared to equivalently trained ResNets. In Figure 4, out of the seven pre-trained models that perform best out-of-distribution six are ViTs. Ablating architecture alone while holding dataset, training augmentations, and parameter count constant, we can compare the model pairs "MoCo. v3, RN" and "MoCo. v3, ViT", "RN-DINO" and "ViT-DINO", and "RN-INSUP" and "ViT-INSUP." In the latter two pairs, the ViT variant is much stronger out-of-distribution than the ResNet variant. For MoCo, the two variants achieve similar performance out-of-distribution.

**Summary.** This section identified which pre-trained models generalize, with several interesting findings. First, models designed for manipulaiton do not necessarily perform well under subtle distribution shifts in comparison to more standard pre-training methods. Second, the presence or absence of supervision does not matter as much as other factors on both in- and out-of-distribution generalization. Finally, ViTs have a slight edge over ResNets in out-of-distribution generalization.

## 5 PROPERTIES OF ROBUST VISUAL REPRESENTATIONS FOR MANIPULATION

Our findings in the last section are both surprising and somewhat unsatisfying because they contradict many of our intuitions about scale and generalization. In our evaluation suite, we saw that better generalization is not cleanly explained by more data, bigger models, or more relevant data. The goal of this section is to identify the properties of pre-trained models that are predictive of generalization. To that end, we correlate out-of-distribution performance with three metrics that have been previously connected to generalization in the machine learning and computer vision literature—in-domain performance, accuracy of a linear probe trained on ImageNet, and shape-bias. We also include a fourth metric, which is specific to ViTs: the emergent segmentation accuracy of the output attention heads. We describe each metric in detail in Section 5.1, discuss our setup for correlating performance in Section 5.2, and analyze our results in Section 5.3.

## 5.1 METRICS

**ID vs OOD.** One of the goals of this paper is to understand how well the findings from existing evaluations of pre-trained models hold under the inevitable environment changes that we expect to see in a real-world setting. If in-distribution performance is reasonably predictive of generalization to our suite of distribution shifts, it is sufficient for researchers to continue developing pre-trained models with existing methods of evaluation. Past work has also shown that the in-distribution performance of a pre-trained model is positively correlated with out-of-distribution performance for a variety of computer vision tasks (Miller et al., 2021). Concretely, we measure in-distribution performance as the success rate of the policy within the training distribution.

**Imagenet vs OOD.** Training linear probes on Imagenet is a common protocol for evaluating the quality of learned representations (He et al., 2019; Chen et al., 2020). Hu et al. (2023) make the related finding that the ImageNet $k$-NN accuracy of a pre-trained model is predictive of performance on imitation learning with a visual reward function. We evaluate ImageNet validation set accuracy for all models with linear probes available.

**Shape-Bias vs OOD.** Shape bias is the extent to which a model makes prediction decisions based on shape. We calculate shape bias as the percent of shape classification decisions out of the set of texture or shape classifications on the Stylized-ImageNet validation set (Geirhos et al., 2019) using the same probes described above.

Training Distribution

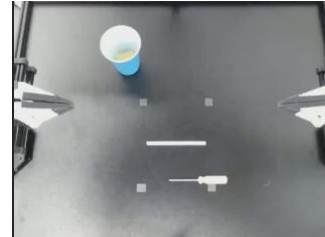

Test Distribution

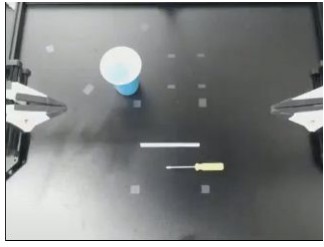

Figure 7: Real world training and test distribution. The test distribution differs from the training distribution in the position of the target objects and the direction of the lighting.

**Jaccard vs OOD.** Finally, for all of the ViT models, we look at the emergent segmentation performance. We denote this nonlinear, deterministic transform as $M$. Formally, we compute the Jaccard index by calculating the mIoU on the PASCAL VOC validation set, $D_{Pascal}$:

$$J(x_i, x_j) = \mathbb{E}_{D_{Pascal}} \left[ \frac{A \cap B}{A \cup B} \right]$$

Where $A$ is a shorthand for positive classification for the target class by $M(\phi(\cdot))$ and $B$ is a shorthand for positive label for the target class. $J$ is evaluated pixel-wise over image indices $x_i$ and $x_j$. We evaluate the Jaccard index of an interpolated attention map averaged across heads in the last attention block at the [CLS] token.

## 5.2 SETUP

We measure the coefficient of determination ($R^2$) and Spearman's rank correlation ($\rho$) for the correlation between the out-of-distribution success rate and each metric described above. Our goal is to find a metric that will result in high correlation between the metric and the OOD success, i.e. both coefficients being close to $1.0$. We fit separate trend lines to ViTs and ResNets. Because of the lack of available probes, we exclude MVP, MVP ViT-S HOI, R3M, VIP, and MAE-IN ViT-S from the shape bias and ImageNet probe correlations. Each point represents one of the 15 pre-trained models we evaluated and represents the average of 6,000 evaluation runs.

## 5.3 RESULTS

We visualize the correlation between each metric and the average out-of-distribution success rate in Figure 5. Although we see a positive relationship between in- and out-of distribution generalization, there are pre-trained models that notably deviate from this trend. Among ViT models one example is MVP, ViT-S (HOI): the average success rate of this model drops to 6.63 from 39.86. By contrast, we find that ImageNet accuracy of a linear probe poorly predicts generalization performance for ViTs.

We also see little correlation between shape-bias and OOD performance for ViT models, but a promisingly strong correlation on the subset of ResNets evaluted. This is surprising because humans make highly shape-biased decisions and increasing shape-bias increases the robustness of imagenet trained CNNs (Geirhos et al., 2019; 2021). One explanation of this finding is that the ViT architecture obviates the need for shape-biased features. For example, a ResNet-50 trained with the DINO training scheme has a strong shape-bias, but not the equivalent ViT model.

Finally, we visualize the relationship between the Jaccard index and OOD performance on all ViT models in Figure 5. There is a strong positive correlation between Jaccard index and OOD performance both in terms of rank correlation and the coefficient of determination. These results suggest that while shape-bias may not be predictive of the OOD generalization ability of a pre-trained ViT, the segmentation ability is a predictive alternative.

| Model | Success |
|---|---|
| MVP | 0% |
| MoCo-v3 | 40% |

Table 1: Success rates on the task of picking up the screwdriver.

One counter-argument to the use of Jaccard index as a metric for for OOD performance is that it would be less predictive for object-level distribution shift, which would occur any time a large distractor is placed in the background of the image. In Figure 6, we plot the success rates of each ViT model as the number of objects increases and verify that the models with the higher Jaccard index actually maintain the highest performance as the number of distractors increases.

## 5.4 VALIDATING IN THE REAL WORLD

In this section, we validate our finding on a real-world generalization scenario by comparing a ViT-B model designed for control (MVP) against a model not designed for control but with a high emergent segmentation score (MoCo-v3).

**Setup.** We learn policies for picking up a screwdriver on the ALOHA setup using the ACT training framework (Zhao et al., 2023). The training dataset is comprised of 50 episodes collected by an expert human demonstrator. Images are collected from 4 camera view points (one on each wrist, one top camera, and one front camera). We replace the standard encoder with a ViT-B and change the initialization of the encoder based on the experimental condition (i.e., we select for a different pre-trained model). We follow the standard ACT training paradigm with the hyperparameters listed in Appendix Table 4. From the training data to the test runs there is a distribution shift in both the placement of the target object (the screwdriver) and in the direction of the lighting. This is visualized in Figure 7. We calculate success on screw pick ups averaged over 10 rollouts in the test environment.

**Results.** We find that MoCo-v3 is stronger on this setting than MVP, even though it is not explicitly designed for manipulation. We find that the MoCo-v3 initialized encoder is able to achieve a success rate of 40% on this task while the MVP initialized encoder is not able to successfully grasp the target object. Qualitatively, the MVP model fails in localizing the object when attempting the grasp, whereas MoCo-v3 model reliably localizes the object, but experiences more failure in finding the right grasp point.

## 6 CONCLUSION

**Summary.** In this paper, we make several surprising findings about the generalization ability of pre-trained visual representations for manipulation tasks. First, we find that, contrary to the current direction in the literature, models pre-trained on manipulation-relevant data do no necessarily generalize better than models trained on standard pre-training datasets (such as ImageNet). Instead, we uncover a recipe for strong generalization: ViT models with a high emergent segmentation accuracy generalize well under visual distribution shifts. Emergent segmentation accuracy is not only a stronger predictor of generalization than many other metrics for robustness, but also requires no additional training to evaluate. This insight can guide the development of pre-trained vision models in future work: preferring architecture development and training algorithms that lead to strong emergent segmentation as opposed to only training on more manipulation-relevant data.

## 7 REPRODUCIBILITY

All of our code is open-sourced and all changes to relevant libraries are available in the supplementary materials. We also include all of the XML files that we used to generate our visual shift scenarios. Our appendix includes the exact hyperparameters we used to conduct our simulated policy training and our real-world experiments.

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

## A APPENDIX

### A.1 PRE-TRAINED MODEL DETAILS

**RN-INSUP** (He et al., 2016) is a ResNet model trained on the ImageNet classificaiton task. We use the default weights and model provided by the Pytorch (Paszke et al., 2019) library.

**ViT-INSUP** is a Vision Transformer (Dosovitskiy et al., 2021) that has been distilled (Touvron et al., 2021) from a larger network that was trained on the ImageNet classification task. In our experiments, we use the model weights and architecture provided in Naseer et al. (2021) with a patch size of 16.

**SIN-SUP** (Naseer et al., 2021) trains a vision transformer on Stylized Image-Net (SIN) (Geirhos et al., 2019). The SIN dataset was constructed to increase the degree to which a model makes predictions on shape instead of texture. Our model weights come from Naseer et al. (2021) and we use the non-distilled DeiT (Touvron et al., 2021) training variant.

**ViT-DINO** (Caron et al., 2021) is trained with extensive augmentations and a self-supervised, contrastive loss that together lead to emergent segmentation within the self-attention heads of the ViT model. We use the model and weights provided by Caron et al. (2021). Interestingly, we don't find the DINO objective to lead to a high shape-bias. This suggests that there are other metrics that measure the degree to which a model is object-centric other than shape-bias.

**ResNet50-DINO** is learned with the same recipe as ViT-DINO. We use the model and weights from Caron et al. (2021).

**MoCo. v3, RN** (Chen* et al., 2021) leverages a contrastive loss with momentum encoding (He et al., 2019) of positive targets. It is trained with the same recipe as MoCo. v3, ViT-B.

**MoCo. v3, ViT-B** (Chen* et al., 2021) are trained in a similar manner as the original MoCo (He et al., 2019), but with changes to improve the stability of training, which are specific to the ViT archieture. We use the checkpoint after 300 epochs.

**MoCo. v3, ViT-S** (Chen* et al., 2021) is trained in a similar manner as MoCo. v3, ViT-B. Even though the smaller model benefits from a longer training horizon, we use the checkpoint at 300 epochs for consistency.

**MAE-IN, ViT-S** follows the same training recipe as MVP, but on top of the ImageNet dataset. We use the weights provided by Radosavovic et al. (2022).

**R3M** (Nair et al., 2022) trains a ResNet model with a combination of manipulation-specific losses–including a time-contrastive loss (Sermanet et al., 2018), video-language alignemnt loss, and L1-regularization–on the Ego4D (Grauman et al., 2022) dataset.

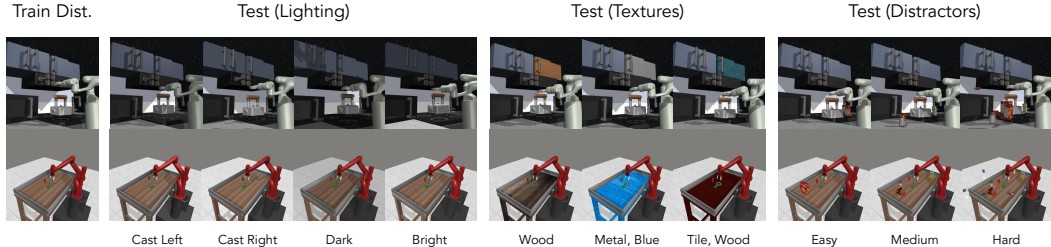

Figure 8: We visualize each distribution shift from the left camera angle on the FrankaKitchen (top) and Meta-World (bottom) environments.

**MVP** (Radosavovic et al., 2022) trains a ViT-B for masked autoencoding (MAE) (He et al., 2021) on the Ego4D (Grauman et al., 2022), Something-Something (Goyal et al., 2017), YouTube 100 Days of Hands (Shan et al., 2020), EpicKitchens (Damen et al., 2018), and ImageNet (Deng et al., 2009) datasets. Unlike R3M, the model is not designed to be exclusive to manipulation.

**MVP, ViT-S (HOI)** (Xiao et al., 2022) is a predecessor of the model described above that trains a ViT-S/16 with an MAE objective on Something-Something (Goyal et al., 2017), YouTube 100 Days of Hands (Shan et al., 2020), EpicKitchens (Damen et al., 2018), and ImageNet (Deng et al., 2009).

**VIP** (Ma et al., 2022) uses an action-free dual of the Algaedice (Nachum et al., 2019) objective to learn representations that are useful for trajectory optimization or reinforcement learning of control tasks unseen during representation pre-training. They train a ResNet-50 on Ego4D with this objective.

**CLIP, ViT-B/16** (Radford et al., 2021) uses contrastive language-image pre-training to learn visual representations trained on an extensive internet datsaet. The learned models exhibit strong zero-shot performance for multiple tasks such as image classification.

**DiNo v2, ViT** (Oquab et al., 2023) scales  Caron et al. (2021) to more parameters and a larger dataset. The full model is a 1B parameter ViT trained on LVD-142M, which is a 142M frame dataset composed of ImageNet-1k, ImageNet-22k, Google Landmarks (Weyand et al., 2020), and a collection of other datasets spanning fine-grained classification, segmentation, depth estimation, and retrieval. The full model is distilled into smaller models. We select the ViT-S distilled model for our experiments. In Table 2, we list the augmentations used on the teacher model. The training loop is only lightly modified during distillation. Suprisingly, the v2 model sees worse in- and out-of-domain performance on our evaluation suite in spite of being distilled from a ladrger model trained on a bigger dataset.

## A.2    DETAILS OF THE ENVIRONMENTS

**FrankaKitchen** (Gupta et al., 2019) is a simulated kitchen environment with a 9-DoF Franka robot. There a multiple household objects available for interaction. The environment is designed to compose tasks together hierarchically, but we focus on learning policies to successfully complete a single task. The episode length is 50 and we inherit the randomization scheme used in R3M, which randomizes the position of the kitchen at the start of each episode.

**Meta-World** (Yu et al., 2019) is a simulated manipulation environment that consists of various table-top manipulation interactions. Unlike FrankaKitchen, the scene objects vary between different tasks. The positions of the objects are randomized at the start of each episode. The maximum episode length is 500.

## A.3    DETAILS OF THE DISRIBUTION SHIFTS

Each distribution shift is visualized from the left camera angle in Figure 8. We don't use the MuJoCo scanned object dataset that is used in (Xie* et al., 2023) because of imperfections in the coloring of the textures.

| Name | Loss Function | Architecture | Datasets | Augmentations |
|---|---|---|---|---|
| RN-INSUP | BCE-Loss | ResNet-50 (23M params) | ImageNet (1.2M frames) | Random crop, Horizontal flip |
| ViT-INSUP | BCE-Loss | ViT-S/16 (22M params) | ImageNet (1.2M frames) | Random crop, Horizontal flip |
| SIN-SUP | BCE-Loss | ViT-S/16 (22M params) | Stylized-ImageNet (1.2M frames) | Random crop, Horizontal flip |
| ResNet50-DINO | Distillation | ResNet-50 (23M params) | ImageNet (1.2M frames) | Multi-crop, Color-jittering, Gaussian blur, Solarization |
| ViT-DINO | Distillation | ViT-S/16 (22M params) | ImageNet (1.2M frames) | Multi-crop, Color-jittering, Gaussian blur, Solarization |
| MoCo. v3, RN | Contrastive | ResNet50 (23M params) | ImageNet (1.2M frames) | Resize, Color-jittering, Horizontal flip, Grayscale, Gaussian blur, Solarization |
| MoCo. v3, ViT-S | Contrastive | ViT-S/16 (22M params) | ImageNet (1.2M frames) | Resize, Color-jittering, Horizontal flip, Grayscale, Gaussian blur, Solarization |
| MoCo. v3, ViT-B | Contrastive | ViT-B/16 (88M params) | ImageNet (1.2M frames) | Resize, Color-jittering, Horizontal flip, Grayscale, Gaussian blur, Solarization |
| MAE-IN, ViT-S | Masked auto-encoding | ViT-S (22M params) | ImageNet (1.2M frames) | Random resize, Random crop |
| R3M | Time-contrastive, L1-regularization, Video-lang alignment | ResNet-50 (23M params) | Ego4D (4.3M frames) | Random crop |
| MVP, ViT-S (HOI) | Masked auto-encoding | ViT-S (22M params) | EpicKitchens 100 Days of Hands, Something-Something (700k frames) | None |
| MVP | Masked auto-encoding | ViT-B (88M params) | Ego4D, ImageNet EpicKitchens, 100 Days of Hands, Something-Something (4.5M frames) | None |
| VIP | Algaedice Dual | ResNet-50 (23M params) | Ego4D (4.3M frames) | Random crop |
| CLIP, ViT-B/16 | Contrastive | ViT-B/16 (88M params) | Internet data (400M pairs) | Random crop |
| DiNo v2, ViT | Distillation | ViT-S/14 (21M params) | LVD (142M frames) | Multi-crop, Color-jittering, Grayscale, Gaussian blur, Solarization |

Table 2: List of pre-trained models with corresponding loss function, augmentations, and datasets used for pre-training. We color code by the data and loss type: ImageNet supervised, self-supervised, trained specifically for manipulation or control tasks, and other.

## A.4 POLICY TRAINING DETAILS

We learn a 2-layer MLP on top of the pre-trained, frozen features with 10 demonstrations. We use the same expert demonstrations as in R3M. We train policies independently over the 'left_cap2' and 'right_cap2' camera angles and show results averaged over both camera angles. We also provide proprioception to the policy. The final performance is averaged over the task settings for each seed. The hyperparamters for policy training are summarized in Table 3. Error bars are 95% confidence interval over seeds.

| Hyperparameter | Value |
|---|---|
| Loss type | MSE |
| Learning rate | 0.001 |
| Batch size | 32 |
| Train steps | 20,000 |
| Optimizer | Adam |

Table 3: Hyperparameters for IL Policy Training

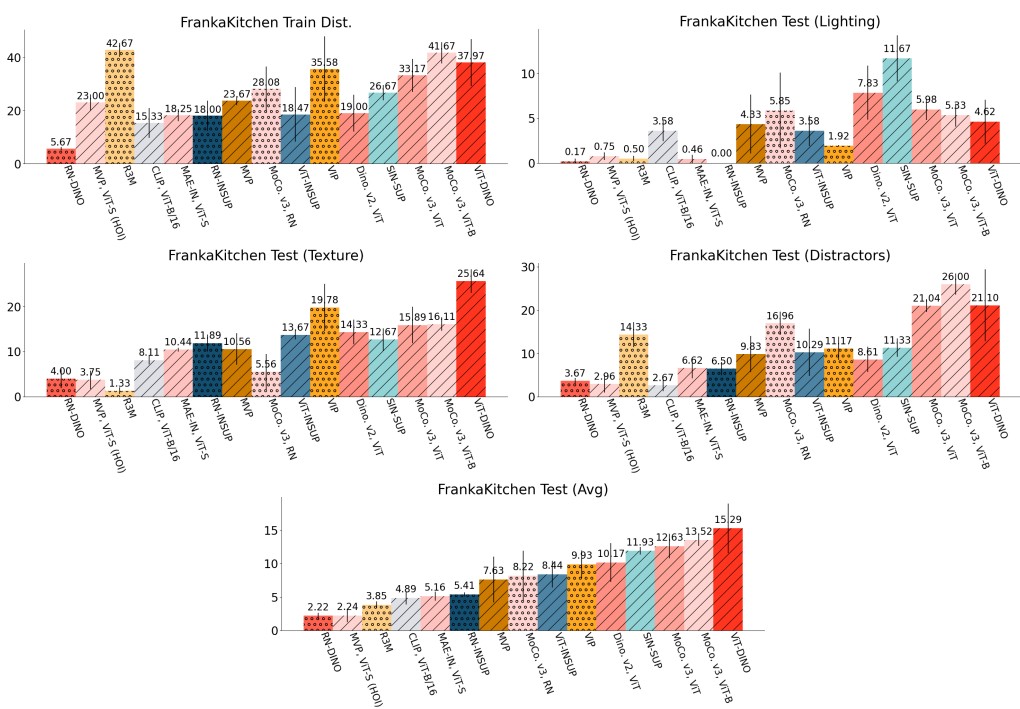

Figure 9: **Detailed OOD Performance on FrankaKitchen.**

## A.5 OOD PERF DETAILS

To provide a more granular understanding of how the complete set of models performs on our evaluation suite, we break down performance by distribution shift type and environment in Figures 9 and 10.

## A.6 IMAGENET VS OOD DETAILS

To evaluate ImageNet accuracy, we use all publicly available probes that have been trained on top of the frozen model features and evaluate them on the ImageNet validation set. The models with available probes are RN-INSUP, RN-DINO, MoCo. v3 RN, ViT-INSUP, ViT-DINO, MoCo. v3 ViT, Dino v2 ViT, MoCo. v3 ViT, SIN-SUP, and CLIP ViT-B/16 and we use the probes that are provided in the implementations cited in Section A.1.

## A.7 SHAPE-BIAS DETAILS

We evaluate shape-bias using the 'model-vs-human' evaluation framework from Geirhos et al. (2021) and use the same probes from Section A.6 to get classification results on the cue-conflict validation dataset ($D_{cue-conflict}$). The cue-conflict dataset contains images where the shape and texture cues are in conflict (e.g., a cat with the texture of the elephant). The shape bias of the model is the ratio of classification decisions made based on the shape cue (e.g., cat) vs the texture cue (e.g., elephant).

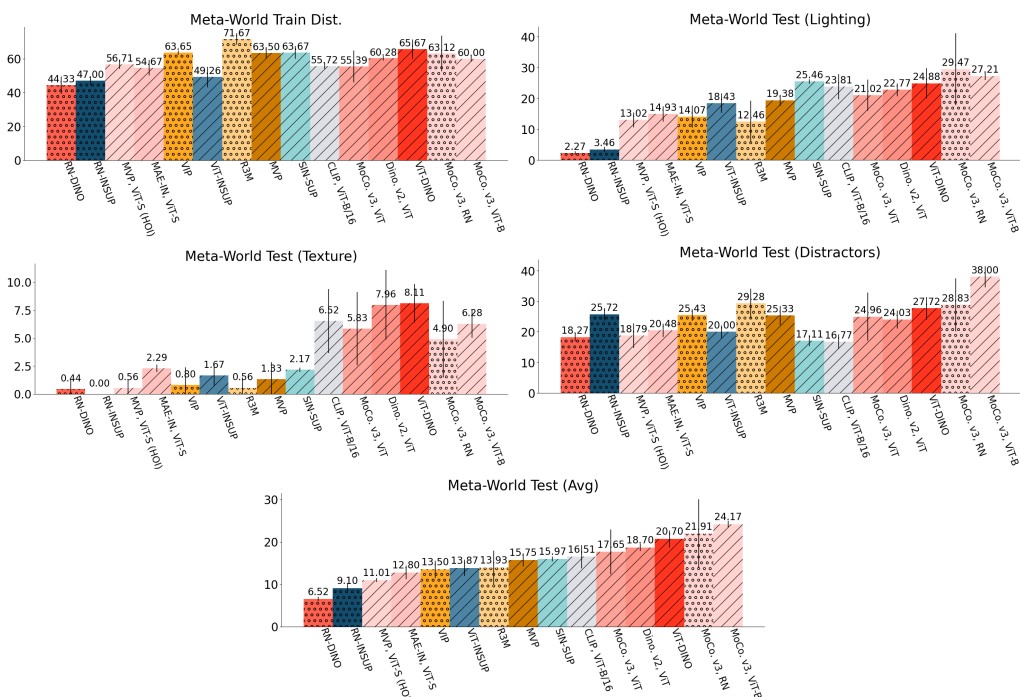

Figure 10: **Detailed OOD Performance on Meta-World.**

Notably, Naseer et al. (2021) find that vision transformers are more shape-biased when making classification decisions than equivalently trained convolutional networks. In our results, we don't find vision transformers to be more strongly shape biased. Vision transformers and convolutional networks vary in how they handle spatial resolution: spatial resolution decreases in each layer of ResNet-50 but remains constant within a ViT. This could explain why we see the ViT architecture somewhat obviating the need for shape-bias in our results.

## A.8 DIFFERENT LEVELS OF DISTRACTORS

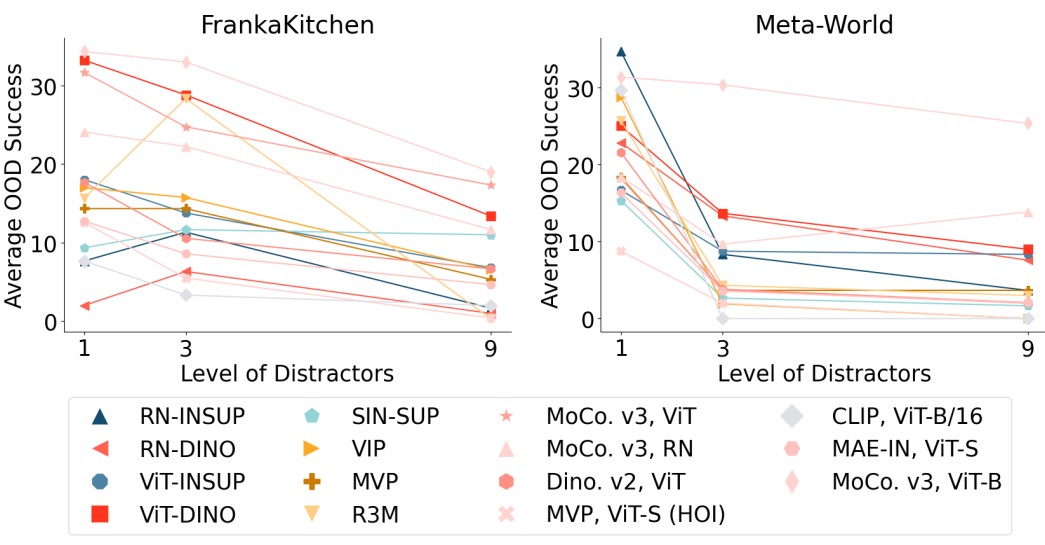

Figure 11: Different levels of distractors.

We extend Figure 6 by including results for ResNets in Figure 11. Models are color coded using the original color scheme in the paper.

## A.9 FINETUNING

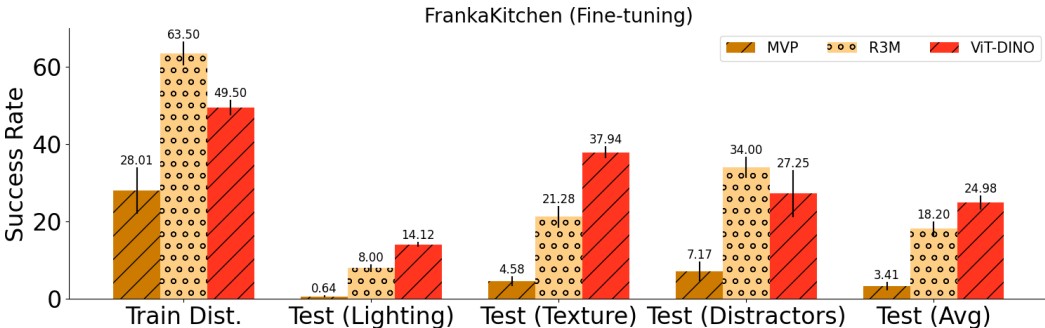

Figure 12: **Finetuning in FrankaKitchen.**

Because the goal of this paper is to probe the quality of learned representations, we follow the tradition of performing evaluation on top of frozen model features. This evaluation is also consistent with the increasing view of pre-trained visual representations as "foundation models" (Bommasani et al., 2022; Oquab et al., 2023) that can be deployed without any gradient updates. Nonetheless, even in the fine-tuning regime, in Figure 12 we still see stronger performance from models that are not designed for manipulation. In this setting, we increased the number of demonstrations to 25 to allow for more data diversity when training the encoders.

## A.10 REAL-WORLD EXPERIMENT DETAILS

Our demonstration data contains two subtasks: an initial screwdriver pick-up and then a handover that happen in sequence. We only evaluate success on the subtask of picking up the screwdriver.

| Hyperparameter | Value |
|---|---|
| Chunk Size | 100 |
| KL Weight | 10 |
| Batch size | 8 |
| Epochs | 10,000 |
| Optimizer | Adam |
| Learning Rate | 1e-5 |

Table 4: Hyperparameters for Policy Training

## A.11 ADDITIONAL EXPERIMENTS

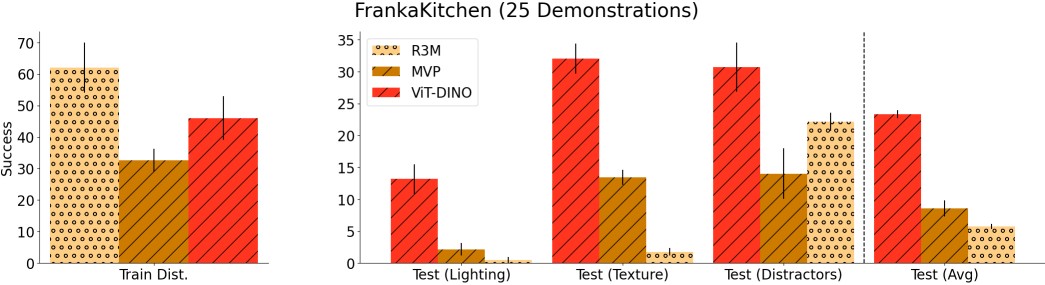

Figure 13: Performance on FrankaKitchen with more demonstrations remains consistent.

**Increasing the number of demonstrations.** To study the impact of increased demonstrations, we also show the performance of R3M, MVP, and ViT-DINO with 25 demonstrations in Figure 13. This is the largest number of demonstrations used in the R3M evaluation suite. We find that the trend remains consistent, if not exaggerated, with increased demonstrations.

**Comparison to PIE-G.** PIE-G Yuan et al. (2022) blends pre-training together with reinforcement learning to learn visually robust representations. The pre-trained model used in PIE-G is a ResNet-18 with features extracted from the second layer of the network. We extract these features for comparison in our benchmark and perform average pooling along the spatial dimensions (that is, along the height and width) to produce a 128-dimensional feature. This model achieves a training performance of 0.0 across all the FrankaKitchen training tasks.

**Analysing the Jaccard index of GradCAM applied to ResNet models.** In our experiments, the Jaccard index was the most predictive metric of out-of-distribution performance. To arrive at an equivalent metric for ResNet models, we evaluate the Jaccard index of segmentation maps generated with Grad-CAM (Selvaraju et al., 2017). To generate our segmentation maps, we use the Grad-CAM implementation made available by Gildenblat & contributors (2021). Figure 14 shows that generating segmentation maps in this way does not give a predictive metric for out-of-distribution performance for ResNets. One explanation for this result is that Grad-CAM is not the best measure of the internal spatial features of a ResNet model. Another hypothesis is that ViTs have the capacity to model shape directly in their attention heads, which obviates the need for shape-biased features. The ResNet model architecture may not have the capacity to support this kind of representaiton, which requires shape bias to be encoded directly in features (Geirhos et al., 2019).

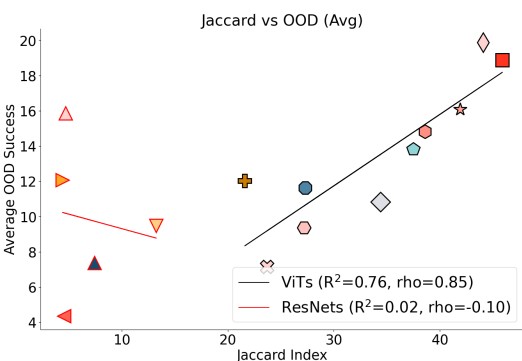

Figure 14: Jaccard index with ResNet models included.

