# OpenReview forum: "What Makes Pre-Trained Visual Representations Successful for Robust Manipulation?"
_ICLR.cc/2024/Conference — Submitted to ICLR 2024_

### Official Review · Reviewer_fmRi · 2023-10-28

**Soundness:** 2 fair
**Presentation:** 3 good
**Contribution:** 2 fair
**Rating:** 5
**Confidence:** 4

**Summary:**

This paper investigates the performance of 15 pre-trained vision models under different visual appearances. It finds that ViT is a strong predictor of out-of-distribution generalization. This paper is to thoroughly answer the questions “which models generalize?” and “how can we predict how well a pre-trained model will generalize?”

**Strengths:**

1.	The paper is well-written and easy to follow. It's worth noting that there is a summary after each experimental section, which aids in clearly grasping the conclusions.

2.	The paper utilizes numerous pre-trained models to substantiate its findings.

3.	The paper includes the real-world experiments.

4.	I believe the conclusions presented can offer some valuable insights to the field.

**Weaknesses:**

1.	The testing scenarios are not significantly different from the experimental settings, leading to conclusions that may not be solid.

2.	The scope of the conclusions is too narrow. While pre-trained models for visuo-motor control is a broad topic, this paper's findings are limited to the relationship between the ViT model and OOD performance under visual distribution shift, which limits in the specific setting and specific model. In this filed, there are too many different conclusions under different settings. Some slight differences may change a lot.

3.	The conclusion that models pre-trained on manipulation-relevant data do not necessarily generalize better than models trained on standard pre-training datasets has been similarly addressed in previous papers.

**Questions:**

1.	"We develop a benchmark for out-of-distribution generalization within FrankaKitchen and Meta-World." I think such a level of modification to the environment doesn't qualify as developing a benchmark. It's more accurate to say that some environments were constructed to verify our conclusions.

2.	I find the spatial features in the figure intriguing. Could you conduct more analyses on how they change with varying levels of distracting factors? Additionally, I would suggest annotating Figure 1 with the value of the Jaccard Index.

3.	Which pre-trained models were trained by you, and which ones are off-the-shelf?

4.	The description for Figure 5 appears twice.

Some related works:

[1] Lin, Xingyu, et al. "SpawnNet: Learning Generalizable Visuomotor Skills from Pre-trained Networks." arXiv preprint arXiv:2307.03567 (2023).

[2] Yuan, Zhecheng, et al. "Pre-trained image encoder for generalizable visual reinforcement learning." Advances in Neural Information Processing Systems 35 (2022): 13022-13037.

**Details Of Ethics Concerns:**

No ethic cocerns.

---

> ### Author Response · Authors · 2023-11-21
> **Response to Reviwer fmRi**
>
> _1. The testing scenarios are not significantly different from the experimental settings, leading to conclusions that may not be solid._
>
> Some visual distribution shifts (such as lighting) are subtle and may only include slight changes in brightness or the direction of shadows. In spite of this, we see that lighting shifts cause the largest average drop in policy performance on FrankaKitchen when compared to more obvious changes such as changes in background texture or the presence of distractors. This suggests that the difficulty of zero-shot transfer may not correlate well with visual difference as perceived by humans.
>
> _2. The scope of the conclusions is too narrow. While pre-trained models for visuo-motor control is a broad topic, this paper's findings are limited to the relationship between the ViT model and OOD performance under visual distribution shift, which limits in the specific setting and specific model. In this filed, there are too many different conclusions under different settings. Some slight differences may change a lot._
>
> One of the striking insights of our paper is the high correlation between the Jaccard index of the attention heads of a ViT model and the out-of-distribution performance of pre-trained models. In addition to this, we include an in-depth comparison of 5 ResNet models and a substantial discussion and analysis of the role of different datasets and pre-training strategies in yielding robust pre-trained models in sections 3 and 4.
>
> Broadly speaking, how best to pre-train visual representations is an open question in the field of learning based robotics. Much of the past work tried to train on datasets like Ego4D that include human-object interactions under the suspicion that these will be better for control. This paper shows that there are other design choices that supersede dataset choice. Moreover, we present an alternative metric to help guide the development of such models going forward. We believe this is a valuable contribution that the ICLR community could benefit from.
>
> _3. The conclusion that models pre-trained on manipulation-relevant data do not necessarily generalize better than models trained on standard pre-training datasets has been similarly addressed in previous papers._
>
> Could you provide a specific reference that shows this? To our knowledge, the only work that shows this is [1], which was made publicly available after our work was submitted to ICLR and is therefore considered concurrent work.
>
> _4. "We develop a benchmark for out-of-distribution generalization within FrankaKitchen and Meta-World." I think such a level of modification to the environment doesn't qualify as developing a benchmark. It's more accurate to say that some environments were constructed to verify our conclusions._
>
> We’ve changed the wording in that paragraph to “We construct environments to study out-of-distribution generalization within FrankaKitchen and Meta-World.” Please find the change highlighted in the updated PDF.
>
> _5. I find the spatial features in the figure intriguing. Could you conduct more analyses on how they change with varying levels of distracting factors?_
>
> Yes, please see Figure 6 and the last paragraph of Section 5.2. From that section:  “One counter-argument to the use of Jaccard index as a metric for OOD performance is that it would be less predictive for object-level distribution shift, which would occur any time a large distractor is placed in the background of the image. In Figure 6, we plot the success rates of each ViT model as the number of objects increases and verify that the models with the higher Jaccard index actually maintain the highest performance as the number of distractors increases.”
>
> _6. Which pre-trained models were trained by you, and which ones are off-the-shelf?_
>
> Consistent with past work on evaluating pre-trained visual representations [2], all of the models we study are off-the shelf.
>
> _7. Additionally, I would suggest annotating Figure 1 with the value of the Jaccard Index. The description for Figure 5 appears twice._
>
> Thank you for pointing these out. Please find both updated in the latest draft.
>
> [1] Dasari et. al. An Unbiased Look at Datasets for Visuo-Motor Pre-Training. https://arxiv.org/abs/2310.09289. CoRL 2023.
> [2] Hu et. al. For pre-trained vision models in motor control, not all policy learning methods are created equal. https://arxiv.org/abs/2304.04591. ICML 2023.

---

> ### Author Response · Authors · 2023-11-22
>
> Hello Reviewer fmRi,
>
> We sincerely appreciate the time you have invested to provide valuable feedback on our paper. We would like to ensure that we have fully addressed your concerns, particularly regarding the scope of our experiments and the design of our experimental protocol. If there are any lingering concerns, please let us know what we could change and we would be happy to take the time to implement. Otherwise,  would you consider raising our score so that we can share the conclusions of our paper more broadly?
>
> Thank you,
> The Authors of Submission 1442

---

### Official Review · Reviewer_ujtD · 2023-10-29

**Soundness:** 2 fair
**Presentation:** 2 fair
**Contribution:** 2 fair
**Rating:** 5
**Confidence:** 4

**Summary:**

The paper presents an empirical evaluation of the efficacy of various pre-trained learning strategies and models in the context of robotic applications using reinforcement learning.
Through the use of pre-trained encoders, the authors employ imitation learning based on expert demonstrations to derive control policies for two distinct robotic manipulation benchmarks.
A comprehensive comparison is made across 15 pre-trained models, considering both their training performances as well as their generalization capabilities when faced with domain shifts.
From different perspectives, the study seeks to discern whether models pre-trained on task-relevant data yield better transfer results than those pre-trained on generic image datasets, such as ImageNet, and whether self-supervised learning offers better generalization than supervised learning. The study also investigates which architectures demonstrate superior transfer capabilities.
Additionally, the authors propose metrics that could potentially predict the generalization performance during unforeseen domain shifts.

**Strengths:**

*   The paper undertakes a comprehensive study, examining a variety of pre-trained models and pre-training strategies, specifically targeting zero-shot out-of-distribution (OOD) generalization. The sheer volume of models tested is laudable, showcasing a thorough investigative approach.
*    The introduction and utilization of the Jaccard index emerges as an innovative and effective method to predict future model generalization. Targeting high Jaccard index scores might emerge as a valuable auxiliary objective in representation learning tasks.
*    One notable observation is that pre-training on a widely-used dataset like ImageNet yields better results than pre-training on task-specific manipulation datasets. This insight is intriguing and opens up avenues for further exploration and understanding.
*    An understated yet crucial observation lies in the comparison of MAE with MoCO/DINO. It becomes evident that data augmentation plays a pivotal role in out-of-distribution generalization. Notably, the top-performing models in Figure 4 heavily employ robust data augmentation techniques. This underscores the pivotal role data augmentation plays, and it would be beneficial for the authors to delve deeper into this aspect in their discussions.

**Weaknesses:**

*  A pivotal reference, [1], detailing state-of-the-art generalization performance, on the DMcontrol suite generalization benchmark, using embeddings from a ResNet's second layer, is absent. It's imperative that the corresponding PIE-G method be included in the benchmark and discussed.
*  The heavy use of acronyms and specific methods without thorough presentation within the main body of the paper hampers readability, especially for non-experts. Introducing them in the related work or at least highlighting that they are elaborated upon in the appendix would have enhanced clarity. Additionally, acronyms in figures should be clearly defined at the onset of the experimental section.
*  The distinction between in-domain and out-domain generalization is blurry. As I discerned, the paper's in-domain generalization seems to pertain to training performance. This conflicts with more conventional uses of the term, such as evaluation on previously unobserved positions during training.
*  The observation that self-supervised learning outperforms supervised learning in yielding generalized features is already discussed in [2], as acknowledged by the authors, making it less of a novelty.
*  The authors' suggestion to use training performance as an indicator of future generalization seems fairly intuitive, especially in scenarios where no distinct strategies were applied to enhance generalization.
*  The paper presents a hypothesis on shape bias correlating linearly with domain shift success for Resnets, but this is supported by merely three data points. This is a thin foundation for drawing substantial conclusions.
*  The Jaccard index experiment is insightful but would benefit greatly from a more thorough description in the main paper rather than relegating it to the appendix.
*  While the appendix treats all MoCo models as analogous to MoCo, the MoCo method should have been introduced properly.

**Minor Comments:**

* The statement, "Hu et al. (2023) shows that model performance is highly sensitive to evaluation" seems somewhat redundant.
* Figure 2 seems superfluous, and its space might be better utilized for succinctly describing the benchmarked methods.
* Chen et al., 2021 references include an extraneous '*'.
* Figure 1: It should clearly indicate which models are visualized, aside from MVP.
* Figure 3: Considering Figure 4's results, the omission of MoCo is puzzling.
 * An inexplicable empty space is present at the top right of page 9.

**References:**
[1] Pre-Trained Image Encoder for Generalizable Visual Reinforcement Learning. Zhecheng Yuan et al. NeurIPS 2022.
[2] The unsurprising effectiveness of pre-trained vision models for control. Simone Parisi et al. ICML 2022.

**Questions:**

Could you discuss on [1] and if possible add it to your benchmark?

Could you provide a similar experiment than the Jaccard index using any attribution method like Grad-Cam [3] or Rise[4] for instance on the Resnets methods?
When you talk on in-domain generalization, where the evaluation scenarios oberved during training?
In the ViT vs Resnets paraggraph, I don't get the sentence "In Figure 6, out of the seven pre-trained models that perform best out-of-distribution six are ViTs." . Aren't all the mentinned models ViTs since we are loccing at the Jaccard index of their attention heads?

**References:**
[1] Pre-Trained Image Encoder for Generalizable Visual Reinforcement Learning. Zhecheng Yuan et al. NeurIPS 2022.
[3]Grad-CAM: Visual Explanations from Deep Networks via Gradient-based Localization. Ramprasaath R. Selvaraju et al. ICCV 2017
[4] RISE: Randomized Input Sampling for Explanation of Black-box Models. Vitali Petsiuk et al. Arxiv 2018

**Details Of Ethics Concerns:**

*   Could you elaborate on the study by Yuan et al. [1]? Would it be possible to incorporate their work into your benchmark for a more comprehensive comparison?

*    The Jaccard index experiment is compelling. Would it be possible to conduct a parallel evaluation using attribution techniques like Grad-CAM [3] or RISE [4], especially focusing on the ResNet architectures? Such an experiment might provide deeper insights into the model's focus and the critical regions of input.

*    When referring to in-domain generalization, were the evaluation scenarios ones that had been observed during the training phase?

*    In the section comparing ViT and ResNets, there's a statement: "In Figure 6, out of the seven pre-trained models that perform best out-of-distribution six are ViTs." This seems contradictory, as the focus is on the Jaccard index of attention heads, implying all mentioned models should be ViTs. Could you elucidate this?

---

> ### Author Response · Authors · 2023-11-21
> **Response to Reviewer ujtD 1/2**
>
> _1. A pivotal reference, [1], detailing state-of-the-art generalization performance, on the DMcontrol suite generalization benchmark, using embeddings from a ResNet's second layer, is absent. It's imperative that the corresponding PIE-G method be included in the benchmark and discussed. … Could you discuss on [1] and if possible add it to your benchmark? … Could you elaborate on the study by Yuan et al. [1]? Would it be possible to incorporate their work into your benchmark for a more comprehensive comparison?_
>
> We added a reference to PIE-G in the related work. Our reason for not including PIE-G in our benchmark is that PIE-G is a method for adapting pre-trained representations with reinforcement learning while our paper focuses on evaluating the quality of pre-trained representations. The problem statement in PIE-G is similar to papers such as PAD [1] and Secant [2], which we also exclude from our evaluation.
>
> The pre-trained model used in PIE-G is a ResNet-18 with features extracted from the second layer of the network. We ran a new experiment where we use the same pre-trained model as PIE-G and extract features from the second layer and perform average pooling along the spatial dimension to produce a 128-dimensional feature. This model achieves a training performance of 0.0 on the FrankaKitchen tasks. We’ve added a new Appendix section (see section A.11 with more detail about how we achieved this result).
>
> _2. The heavy use of acronyms and specific methods without thorough presentation within the main body of the paper hampers readability, especially for non-experts. Introducing them in the related work or at least highlighting that they are elaborated upon in the appendix would have enhanced clarity. Additionally, acronyms in figures should be clearly defined at the onset of the experimental section._
>
> Due to space constraints, we relegated a detailed description of each model to the Appendix. Based on your suggestions, we’ve improved section 4 to include more direct references to the appendix and earlier definitions of models. Please see the pdf where this update is highlighted.
>
> _3. The distinction between in-domain and out-domain generalization is blurry. As I discerned, the paper's in-domain generalization seems to pertain to training performance. This conflicts with more conventional uses of the term, such as evaluation on previously unobserved positions during training. … When you talk on in-domain generalization, where the evaluation scenarios oberved during training? … When referring to in-domain generalization, were the evaluation scenarios ones that had been observed during the training phase?_
>
> Our in-domain generalization is not training performance. We run learned policies on kitchen and table positions sampled from the training distribution. The scene initializations are not seen during training. In the uploaded PDF, we’ve highlighted this distinction under the “Distribution Shifts” section.
>
> _4. The observation that self-supervised learning outperforms supervised learning in yielding generalized features is already discussed in [2], as acknowledged by the authors, making it less of a novelty._
>
> Could you clarify where in the paper we make this claim? Our results show that the choice to use supervision or not does not matter in comparison to metrics such as the Jaccard index. In paragraph 5 of section 4 we directly cite [2] stating how our results are a counterexample to theirs: “Parisi et al. (2022) also find that self-supervised learning outperforms supervised learning. Our results contradict this finding. Figure 4 shows that supervised training on Stylized ImageNet achieves a higher success rate in the training distribution than self-supervised training on ImageNet with a masked auto-encoding loss. These models maintain the same rank out-of-domain as well.”
>
> _5. The paper presents a hypothesis on shape bias correlating linearly with domain shift success for Resnets, but this is supported by merely three data points. This is a thin foundation for drawing substantial conclusions._
>
> In parts of the paper we make reference to [1] which shows that ImageNet trained CNNs are biased towards texture and that increasing shape bias improves accuracy and robustness. Our goal is not to make any new claims about shape-bias in this paper. If there are sections where we claim this please let us know which ones and we will modify them accordingly.
>
> _6. The Jaccard index experiment is insightful but would benefit greatly from a more thorough description in the main paper rather than relegating it to the appendix._
>
> We’ve moved some of the Appendix in the description into the main paper. Please find the update highlighted in red in Section 5.1.

---

> ### Author Response · Authors · 2023-11-21
> **Response to Reviewer ujtD 2/2**
>
> _8. Could you provide a similar experiment than the Jaccard index using any attribution method like Grad-Cam [3] or Rise[4] for instance on the Resnets methods? The Jaccard index experiment is compelling. Would it be possible to conduct a parallel evaluation using attribution techniques like Grad-CAM [3] or RISE [4], especially focusing on the ResNet architectures? Such an experiment might provide deeper insights into the model's focus and the critical regions of input._
>
> Based on your suggestion, we performed the same analysis with Grad-CAM on the ResNet models and find that there isn’t a strong correlation. Please see the updated appendix section A.11 for more details. Other work finds that robustness in CNNs is correlated with shape-bias as evaluated by accuracy on the cue-conflict dataset. Our leading hypothesis is that shape-bias is the best measure of ResNets for any kind of out-of-distribution performance (again, based on [1]) whereas ViTs have the capacity to model shape directly in the attention heads, which obviates the need for shape-biased features.
>
> _9. In the ViT vs Resnets paraggraph, I don't get the sentence "In Figure 6, out of the seven pre-trained models that perform best out-of-distribution six are ViTs." . Aren't all the mentinned models ViTs since we are loccing at the Jaccard index of their attention heads? … In the section comparing ViT and ResNets, there's a statement: "In Figure 6, out of the seven pre-trained models that perform best out-of-distribution six are ViTs." This seems contradictory, as the focus is on the Jaccard index of attention heads, implying all mentioned models should be ViTs. Could you elucidate this?_
>
> You’re correct: that line should refer to Figure 4 where we compare ViTs and ResNets. Of the 7 best performing models in the Test Distributions, 6 are ViTs. We’ve corrected this mistake in the revised PDF. Thank you for identifying this error.
>
> _Minor comments:_
>
> Please see the revised PDF. Changes are highlighted in red. Thank you for noting these changes.
>
> [1] Geirhos et. al. ImageNet-trained CNNs are biased towards texture; increasing shape bias improves accuracy and robustness. ICLR 2019. https://arxiv.org/abs/1811.12231.

---

> ### Author Response · Authors · 2023-11-22
>
> Hi reviewer ujtD,
>
> We greatly appreciate the time that you have dedicated to reviewing our work. Based on your suggestion, we have improved the paper by adding additional references to and experiments utilizing PIE-G. We have also attempted revise the work to be more clear about the distributions and models studied. Please let us know if there are any unresolved issues that we can take the time to address. If not, would you consider raising your score so that we can increase the visibility of our findings?
>
> Sincerely,
> The Authors of Submission 1442

---

> ### Comment · Reviewer_ujtD · 2023-11-23
>
> I acknowledge the authors' responses to my questions. In particular, they made a clear effort to address my concerns about PIE-G and Grad-Cam. Unfortunately, the results using Grad-Cam on the Jaccard index are somewhat deceptive, suggesting that this indicator is better suited for ViTs. Given this additional information and the clarifications provided by the authors, along with the corrections and additions made to the paper, I am willing to raise my score from 3 to 5.

---

### Official Review · Reviewer_5UTX · 2023-11-01

**Soundness:** 2 fair
**Presentation:** 4 excellent
**Contribution:** 2 fair
**Rating:** 5
**Confidence:** 3

**Summary:**

This work studies the comparative generalisation performance of policies trained on top of pre-trained representations trained for different tasks like control and self supervised learning. The generalisation is studied across visual distribution shifts like texture, lighting and object distractor changes, after training policies on a downstream manipulation task on as few as 10 demos. Experiments show (perhaps counterintuitively) that representations that were trained for control are less robust to the shifts as compared to representations trained with self supervised objectives on imagenet data. They study the performance trends for 2 architectures (ViT vs resNet), and attempt to find a different metric that might be correlated with performance under distribution shift. Among the tracked metrics (shape bias, segmentation ability, linear probing accuracy), they show that segmentation ability of the attention layers of a pre-trained representation model are most predictive of performance of the trained policy after the lighting etc is slightly shifted.

**Strengths:**

1) The premise of finding metrics that could be correlated with downstream robustness is an interesting and a useful direction
2) The study is conducted systematically over a range of architectures and representations.

**Weaknesses:**

1) My main concern is that the paper assumes an artificially restricted setting with unrealistically limited demonstrations for a task (10, collected without varying a single thing in the background like lighting etc), on which even the training distribution performance is highly suboptimal (success rate is 40-60%).  At this performance level, the policy would not be deployed even in the training environment. Typically, in such a limited demonstration regime, the demonstrations would be collected by varying placement and lighting conditions.
I’m not sure (yet) that the paper actually measures what it says it measures - since the demos are so few and the training distribution policy performance is so low, the current degradation in performance under visual shift might not even be due to the representation being bad, but due to the policy network overfitting on correlations in the training data.
2) Consider this: It seems possible to me that if two representations A and B are tested using the paper’s protocol and A encodes some extra variables that are spuriously correlated to actions in the limited demo data, then the policy could latch onto them to identify actions for prediction. However, these extra variables could be useful for prediction over a wider distribution of envs/conditions and if the demos weren’t spuriously correlated with these extra encoded variables, then the learnt policy might have been better than the one learnt using B. Therefore this trend might change if you went from 10 to say 40 demos where in the 40 demos the lightning or another condition is varied in a limited range [x-y] and the test time condition is a value outside this range. Including experiments of this format would make a much more convincing case IMO, and would be much closer to an actual real world use case.

**Questions:**

1) The current description of how the shape bias is calculated is unclear, it would be great if this can be described more explicitly, along with including a motivation for why is it reasonable to adopt this as a metric.
2) Why does Section 5.2 say the probe (and shape bias value) is unavailable for MVP/R3M/VIP models? A linear probe just has to be trained on imagenet on top of the representation right?

---

> ### Author Response · Authors · 2023-11-21
> **Response to Reviewer 5UTX 1/2**
>
> _1. My main concern is that the paper assumes an artificially restricted setting with unrealistically limited demonstrations for a task (10, collected without varying a single thing in the background like lighting etc), on which even the training distribution performance is highly suboptimal (success rate is 40-60%). At this performance level, the policy would not be deployed even in the training environment. Typically, in such a limited demonstration regime, the demonstrations would be collected by varying placement and lighting conditions. I’m not sure (yet) that the paper actually measures what it says it measures - since the demos are so few and the training distribution policy performance is so low, the current degradation in performance under visual shift might not even be due to the representation being bad, but due to the policy network overfitting on correlations in the training data._
>
> _Consider this: It seems possible to me that if two representations A and B are tested using the paper’s protocol and A encodes some extra variables that are spuriously correlated to actions in the limited demo data, then the policy could latch onto them to identify actions for prediction. However, these extra variables could be useful for prediction over a wider distribution of envs/conditions and if the demos weren’t spuriously correlated with these extra encoded variables, then the learnt policy might have been better than the one learnt using B. Therefore this trend might change if you went from 10 to say 40 demos where in the 40 demos the lightning or another condition is varied in a limited range [x-y] and the test time condition is a value outside this range. Including experiments of this format would make a much more convincing case IMO, and would be much closer to an actual real world use case._
>
> Our training protocol is identical to the protocol used to evaluate pre-trained models for manipulation. In particular, we borrow training environments, experiment parameters, and code directly from R3M and recover similar performance within the train distribution as the original R3M does.  This evaluation protocol closely resembles the protocol typically used to evaluate many of the pre-trained models for manipulation.
>
> With that said, we completely agree with the reviewer that the old way of evaluating models (that is, training with demonstrations in a single visual environment and evaluating performance in that environment) is suboptimal. One of the goals of this work is to support that idea.
>
> However, the promise of visual pre-training is to amortize the cost of collecting visually diverse demonstrations. The ideal pre-trained visual representation would enable a researcher to collect a handful of demos in the lab without having to vary the appearance of the background or the direction of the lighting. To use the reviewer's example, we would hope that the pre-trained representation would be able to produce features that are invariant to the feature A that is spuriously correlated with actions. Motivated by this, we chose to _evaluate_ on visually diverse scenes instead of training on them so that we can see how well these models are holding this promise.
>
> To address the reviewer’s concern about the number of demonstrations, we added additional experiments with 25 demonstrations for ResNet50-insup, ViT-DINO, and MVP on FrankaKitchen and find that the same trend holds strongly: ViT models with attention heads that have a high Jaccard index outperform models designed for manipulation. Please find these new results in Appendix Section A.12.
>
> _2. The current description of how the shape bias is calculated is unclear, it would be great if this can be described more explicitly, along with including a motivation for why is it reasonable to adopt this as a metric._
>
> We calculate shape bias in the same way as [1]. Concretely, we take the pre-trained model and a linear probe trained on ImageNet and evaluate the performance of this classifier on the cue-conflict dataset. The cue-conflict dataset contains images where the shape and texture cues are in conflict (e.g., a cat with the texture of the elephant). The shape bias of the model is the ratio of classification decisions made based on the shape cue (e.g., cat) vs the texture cue (e.g., elephant). Shape-bias is used as an evaluation of model quality in many seminal works on computer vision [2,3,4]. We’ve updated section A.7 to describe this in detail.
>
> _3. Why does Section 5.2 say the probe (and shape bias value) is unavailable for MVP/R3M/VIP models? A linear probe just has to be trained on imagenet on top of the representation right?_
>
> Yes. Our goal in this section is to show that these models are not predictive which is clear with the current evaluation set.

---

> ### Author Response · Authors · 2023-11-21
> **Response to Reviewer 5UTX 2/2**
>
> [1] Geirhos et. al. ImageNet-trained CNNs are biased towards texture; increasing shape bias improves accuracy and robustness. ICLR 2019. https://arxiv.org/abs/1811.12231.
> [2] Dehghani et. al. Scaling Vision Transformers to 22 Billion Parameters. ICML 2023. https://proceedings.mlr.press/v202/dehghani23a.html.
> [3] Hendrycks et. al. Natural Adversarial Examples. CVPR 2021. https://openaccess.thecvf.com/content/CVPR2021/html/Hendrycks_Natural_Adversarial_Examples_CVPR_2021_paper.html
> [4] Darcet et. al. Vision Transformers Need Registers. 2023. https://arxiv.org/pdf/2309.16588.pdf.

---

> ### Author Response · Authors · 2023-11-22
>
> Hi reviewer 5UTX,
>
> Thank you again for your feedback on our work. Your recommendations undoubtedly improved the quality of our submission. We are checking in to see if you feel that we have adequately addressed your concerns about the restrictiveness of the training protocol. If you feel they have been addressed, we would greatly appreciate if you could revise your score so that we can share these findings with the ICLR community.
>
> Sincerely,
> The Authors of Submission 1442

---

### Author Response · Authors · 2023-11-22

Thank you again to all of the reviewers for providing suggestions that have undoubtedly improved the quality of our submission. Based on your feedback we have made the following major changes:

- Added results with an increased number of demonstrations for R3M, MVP, and DINO (Section A.11).
- Added reference to and comparisons against the pre-trained model in PIE-G (Section 2 and A.11).
- Evaluated the Jaccard index of Grad-CAM maps from ResNet models as a predictive measures of OOD performance. As expected, we find that these are not predictive. This opens up interesting questions for future work such as: do ViTs and ResNets encode invariant information about shape differently? Do ResNet models lack the capacity to represent shape information within their spatial feature maps unlike ViTs? Is shape-bias uncorrelated with ViT robustness on standard OOD benchmarks? (Section A.11)
- Included more information about Jaccard index in the main paper (Section 5.1).

These and other minor changes are highlighted in red in our revised draft.

__We sincerely hope that these changes have addressed reviewer concerns. If so, we kindly request that you consider revising your scores. To quote the reviewers, this work introduces "an innovative and effective method to predict future model generalization" and makes a "notable observation ... that pre-training on a widely-used dataset like ImageNet [can yield] better results than pre-training on task-specific manipulation datasets." (ujtD) This study is "conducted systematically over a range of architectures and representations" (5UTX), "is well-written and easy to follow", and  "the conclusions presented can offer some valuable insights to the field." (fmRi)__

---

### Meta-Review · Area_Chair_awGU · 2023-12-09

**Metareview:**

**Summary of the Paper**:
The paper examines the effectiveness of various pre-trained vision models in robotic applications under visual distribution shifts, like lighting and texture changes. Focusing on how well these models adapt to manipulation tasks with limited demonstrations, it compares 15 pre-trained models, particularly investigating models trained on task-specific datasets versus those trained on generic datasets like ImageNet. A key finding is that **models with high emergent segmentation abilities**, particularly ViT models, show strong out-of-distribution generalization. The study **introduces the Jaccard index** as a predictive metric for model robustness in such tasks.

**Strengths**:
The paper's comprehensive approach in evaluating a wide range of pre-trained models stands out, providing a detailed analysis of their generalization capabilities. The introduction of the **Jaccard index** as a metric for predicting model robustness in visual distribution shifts is a contribution. Another strength could be the counterintuitive finding that **models pre-trained on generic datasets might outperform those trained on manipulation-specific data**. The systematic experimentation and clear presentation of results contribute to the paper's robustness and readability.

**Weaknesses**:
A limitation is its somewhat **narrow focus on the relationship between ViT models and out-of-distribution performance**, potentially overlooking broader aspects of pre-trained models for visuo-motor control. The experimental settings, with **limited demonstration variations**, might not fully reflect real-world applicability. Some conclusions, like the superiority of ImageNet-trained models over task-specific ones, have been **previously addressed**, reducing the novelty. Additionally, the **reliance on specific architectures like ViT** might limit the generalizability of the findings.

**Justification For Why Not Higher Score:**

It is a paper claiming bold and "surprising" claims. Therefore, clear evidence, experiment protocol, and exposition are crucial. However, none of the reviewers were convinced enough in terms of the above criteria.

**Justification For Why Not Lower Score:**

N/A

---

### Decision · Program_Chairs · 2024-01-16

Reject